# Unsupervised image-to-video domain adaptation for fine-grained video understanding

## Abstract

Video understanding models continue to rely on image-pretrained semantic representations due to a lack of labeled videos. Pixel-precise video annotations are time-consuming and laborious to collect, and may not be feasibly obtained in certain situations. There is a growing amount of freely available unlabeled video data that has led many methods to tackle unsupervised video representation learning and image-to-video domain adaptation. The focus thus far has been on semantic representations for classification, which lack the spatial detail required for tasks such as segmentation. To produce representations better suited for fine-grained video understanding, we propose using large-scale image segmentation datasets and domain adversarial learning to train 2D/3D networks for video segmentation. We introduce a novel unsupervised clustered adversarial loss that first clusters feature maps from a patch embedding then applies a domain discriminator to samples within clusters. Our loss is designed to prevent removal of overall spatial structure while encouraging the removal of fine-grained spatial information specific to the image and video domains. Through experiments using several image and video segmentation datasets, we show how a general or clustered adversarial loss placed at various locations within the network can make spatial feature representations invariant to these domains and improve performance when the network has access to only labeled images and unlabeled videos.

## 1 Introduction

Using large-scale image datasets to pretrain video understanding models is a common method of learning semantic information when there are no video labels available. This can result in subpar performance on downstream video tasks, especially tasks that require pixel-precise localization of objects in the frame. The performance drop observed when training on images and testing on videos is due to the image-video domain gap, which arises due to video artifacts not present in images, namely motion blur, low lighting and low resolution. Figure 1 shows an example of the domain differences arising in common datasets: the boundaries of the moving bicycle in the video frame are blurred. Other contributions to the domain gap are the different distributions of spatial locations of objects in the frame, diversity of object appearance and aspects, and camera framing (Kalogeiton et al., 2015). These factors combined make supervised training on images insufficient for pixel-wise video understanding, producing a need for an alternative representation learning method that uses unlabeled videos.

Unsupervised video domain adaptation has mainly focused on the classification task rather than on segmentation. Recent methods minimize distance between augmented versions of video clips (Li et al., 2023) and utilize spatial (Zara et al., 2023a) and text (Zara et al., 2023b) features from CLIP-pretrained models as a source of pseudo labels for the target domain. (Lo et al., 2023) focus on video segmentation but rely heavily on optical flow, which can be time-consuming to extract. Recent works do not explicitly address the image-video domain gap.

In this paper, we propose an approach to image-to-video domain adaptation that takes advantage of both labeled images and unlabeled videos and apply it to video segmentation. Taking inspiration from Tang et al. (2012), we use unlabeled videos to minimize the domain difference between image representations and the spatial component of video representations. To prevent the loss of discriminative semantic information, we propose a novel clustered adversarial loss in which features

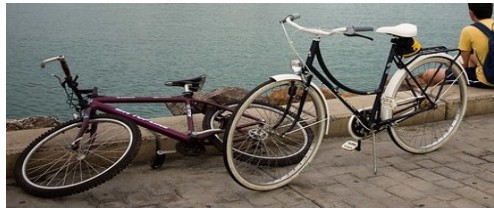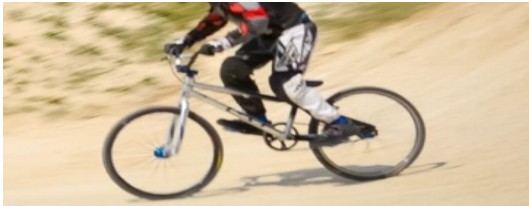

Figure 1: A single image from the COCO dataset (left) and a frame from Davis 2019 (right) shows domain differences such as motion blur arising from video artifacts.

from images and videos are first clustered in an unsupervised manner before applying an adversarial loss to remove domain-specific information from features within each cluster. We train our video segmentation networks to be invariant to properties specific to video (motion blur, viewpoints, etc.) so that we can train on labeled images and apply them to videos without a performance drop.

We present experiments in which our proposed unsupervised loss is combined with an image-supervised segmentation loss, leading to a training method that relies only on image labels but maintains good performance on videos. We experiment with two different segmentation backbones: convolutional neural networks (CNNs) and Transformers. To take advantage of temporal information in unlabeled videos while retaining spatial information from labeled images, we also apply our method to VideoSwin (Liu et al., 2021b) with a spatiotemporal window size. We experiment with two different placements for the domain discriminator – after the patch embedding layer (spatial) or after the encoder (global) – and find that the spatial placement boosts the contribution of the adversarial loss. We conduct experiments using the video segmentation datasets Davis 2019 and FBMS and show that in our target setting with no access to labeled videos, our method improves segmentation performance over models supervised with images. In an ablation where we replace the video dataset with another image dataset, we show that the adversarial loss is indeed removing image-video domain-specific differences from the representations.

## 2 RELATED WORK

**Image to Video Adaptation**   Image-to-video domain adaptation has been explored for video classification (Chen et al., 2021; Yao et al., 2015), detection (Prest et al., 2012; Donahue et al., 2013) and face recognition (Sohn et al., 2017). Kae & Song (2020) propose a two-stage training approach in which 2D image features are transferred to a 3D CNN before continuing to train on videos. Recently, Lin et al. (2022) introduced CycDA, which performs spatial feature learning and spatiotemporal feature learning alternately in a four-stage training approach. There is a lack of image-to-video domain adaptation for video segmentation, where it is important for features to maintain a high spatial resolution throughout the encoding stage. As a result, video segmentation features are more dependent on spatial localization than they are for the tasks above. The adaptation methods developed for those tasks do not take detailed spatial domain differences into account.

Many works have addressed domain adaptation for semantic segmentation in order to solve the synthetic-to-real problem or transfer representations. Shin et al. (2021) use distillation to tackle synthetic-to-real domain adaptation for video semantic segmentation. Guan et al. (2021) enforce similar temporal consistency between consecutive real frames and consecutive synthetic frames. Hong et al. (2017) use unlabeled videos for image segmentation by generating pseudo-labels. Tang et al. (2012) address image-to-video domain adaptation in object detection by retraining an image-trained detector on target video samples. Our clustered adversarial loss is most similar to that of Yang et al. (2021), which uses domain adversarial learning on the class tokens of a vision transformer for image classification.

**Unified Image-Video Models**   A growing body of work is focused on developing unified models for image and video tasks. Huang et al. (2023) propose training the same model for many image and video tasks interactively, but rely on pixel-wise video annotations. Qing et al. (2023) combines spatial features from a frozen CLIP-pretrained transformer with temporal features from a 3D encoder by finetuning temporal and integration branches; their method does not remove image domain-

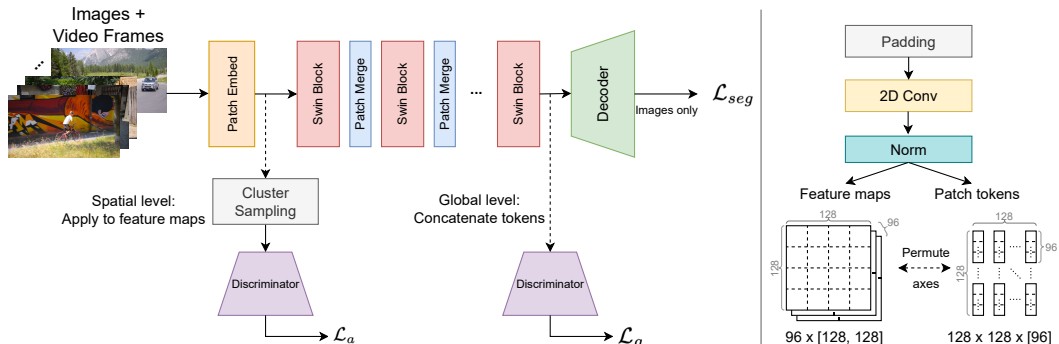

Figure 2: We apply the supervised segmentation loss to images only, and add an adversarial loss on either the spatial or the global level. In the spatial treatment, a discriminator is applied to spatial feature maps following *cluster sampling* (see Fig. 3) the patch embedding layer. In the global treatment, a discriminator is applied to full image representations following the end of the encoder. The patch embedding, shown right, consists of 96 4×4 convolution filters and normalization.

specific features from the spatial representations. Vi2CLR (Diba et al., 2021) uses separate 3D and 2D CNN encoders and contrastive learning over clusters of latent frame and clip features.

## 3 METHODS

Our approach relies on treating images and videos as separate domains and applying a domain adversarial loss to learn robust semantic representations. Many image and video segmentation networks use encoders that learn low-level features in the shallow layers and abstract features in deeper layers. Based on observations by Kalogeiton et al. (2015), we hypothesize that the semantic representations learned by an encoder trained only on images differs from those learned on videos due to video artifacts like motion blur, camera framing and aspect diversity. We further hypothesize the artifacts make it more difficult for a segmentation decoder to recover fine-grained details in videos. Our goal is to bring detailed semantic features learned from images closer to those learned from videos using an adversarial loss propagated through the encoder. In a real-world scenario the most likely setting is access to a small amount of labeled images (and perhaps labeled videos) and a large amount of unlabeled videos. Our method makes it straightforward to use all available data.

To make the feature representations invariant to the image and video domains, we add a domain discriminator at different points in the encoder, shown in Figure 2. During training we reverse the discriminator's gradients when they meet the encoder, which encourages the encoder to learn feature representations that fool the discriminator by containing as little information as possible about the domain.

### 3.1 DOMAIN ADVERSARIAL LOSS

Formally, we are given samples from a source dataset $D^s = \{(x^s, y^s)\}_{s=1}^N$ where each $x^s$ is a single image and $y^s$ is its pixel-wise segmentation label. In the unsupervised domain adaptation setting we have access to samples from an unlabeled target dataset $D^t = \{x^t\}_{t=1}^M$ where each $x^t$ is a video frame. Given a sample $x_i \in \{D^s, D^t\}$, a discriminator estimates its domain $D_i$ through:

$$\underset{D \in \{D^s, D^t\}}{\arg\max} \ \Pr(D_i = D \mid h(x_i)) \tag{1}$$

where $D_i$ is the domain of $x_i$ and $h(\cdot)$ is a function mapping $x_i$ to some latent feature space. In our setting, $h(x_i)$ is the encoder's feature representation of $x_i$ (global level) or a feature map from its low-level embedding (spatial level); Figure 2 shows these placements in the model architecture. Using a binary cross entropy loss, the discriminator's training objective $\mathcal{L}_g$ is to minimize:

$$\mathcal{L}_g = \sum_{(x_i, d_i) \in D} -d_i \log(p_i) + (1 - d_i) \log(1 - p_i) \tag{2}$$

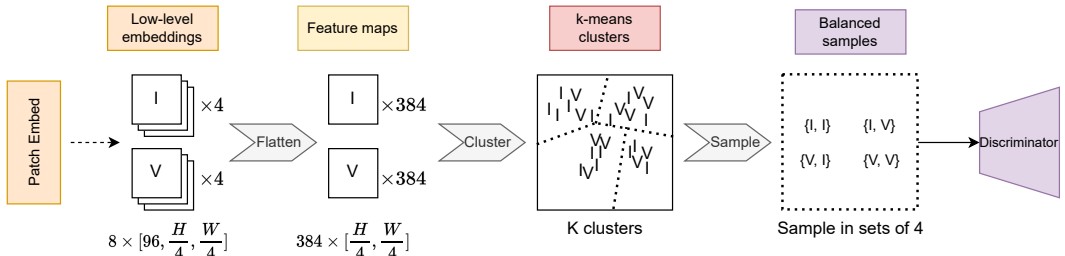

Figure 3: In the clustered adversarial loss feature maps from a batch of low-level image / video frame embeddings are clustered, then balanced samples are drawn from each cluster. *I* represents feature maps derived from images, and *V* represents feature maps derived from video frames. Here the input has dimensions $(H, W)$; there are 96 convolution filters of size $4 \times 4$, and 4 image and 4 video frame samples per minibatch.

where $D = D^s \cup D^t$, $d_i$ is a binary indicator $\mathbb{1}[x_i \in D^s]$, and $p_i$ is the network's estimate of $\Pr(D_i = D^s | h(x_i))$.

During training, a Gradient Reversal Layer (GRL) (Ganin et al., 2016) is placed between discriminator and layer prior to it. The GRL outputs the identity on the forward pass and reverses the sign of the discriminator's gradients during the backward pass.

## 3.2 CLUSTERED ADVERSARIAL LOSS

In our setting, we consider images and videos to be two separate domains, and are interested in removing image- and video-specific information from semantic feature representations. We hypothesize that this will allow the network to learn semantic features that perform well on the target domain, videos, after training on labeled source images. The general adversarial loss $\mathcal{L}_g$ defined in equation 2 has been widely used in domain generalization tasks to remove domain-specific information from features (Lin et al., 2022; Guo et al., 2023). However, it does not spare semantic and spatial structure relevant to the task. This becomes an issue when the source and target domains have very different spatial or class distributions, in which case the discriminator may remove crucial semantic class information from the network, leading to poor performance. Image and video datasets may be labeled with different semantic classes, but we seek to take advantage of all labeled data regardless of annotation consistency.

To prevent this loss of structure, we propose a novel unsupervised clustered adversarial loss that retains any coarse semantic and spatial information that may correlate with domain while still removing fine domain-specific features. To this end we perform unsupervised cluster sampling, shown in Figure 3. Specifically, feature maps extracted from image and video frames are first clustered in an unsupervised manner using $K$-means clustering, then an adversarial loss is applied to balanced samples drawn from each cluster. The discriminator now takes a pair of purely spatial feature maps and estimate the likelihood that they belong to the same domain. Formally, our clustered adversarial loss $\mathcal{L}_{cl}$ is defined as:

$$\mathcal{L}_{cl} = \sum_{k=1}^{k \leq K} \sum_{(x_i, x_i') \in B_k} -b_i \log(q_i) + (1 - b_i) \log(1 - q_i) \tag{3}$$

where $K$ is the total number of clusters, $b_i$ is a binary indicator of whether $x_i$ and $x_i'$ belong to the same domain $D_i$, and $q_i$ is the network's estimated probability that $x_i$ and $x_i'$ belong to the same domain. $B_k$ is a set of pairs of image and video features within a cluster, constructed to contain a balanced number of feature pairs from $(D^s, D^t)$, $(D^s, D^s)$ and $(D^t, D^t)$ in order to stabilize the discriminator's loss. Clustering is only carried out during training and does not affect inference time.

## 3.3 SEGMENTATION LOSS

To maintain detailed semantic information, the network is trained for segmentation through a standard cross entropy loss:

$$\mathcal{L}_{seg} = \sum_{(x_i,y_i) \in D^s} \sum_j \sum_k y_{i,j} \log p_{i,j,k} \tag{4}$$

where $j$ iterates over each pixel in the image, $c$ ranges over the number of object classes and $p_{i,j,k}$ is the decoder's estimate of the probability that pixel $j$ in image $i$ belongs to class $k$. In case of spatiotemporal input, $j$ ranges over the temporal dimension as well.

The final network is trained end-to-end by minimizing the segmentation loss and maximizing the adversarial loss:

$$\mathcal{L}_{final} = \mathcal{L}_{seg} - \mu \mathcal{L}_{\theta_a} \tag{5}$$

where $\mathcal{L}_a \in \{\mathcal{L}_g, \mathcal{L}_{cl}\}$ and $\mu$ is a constant scaling factor chosen through grid search to ensure the encoder backbone does not collapse. Our joint training procedure ensures that the network learns feature representations that are useful for segmentation and do not contain domain information.

# 4 EXPERIMENT SETTINGS

## 4.1 DATASETS

To understand how our method changes semantic segmentation representations learned from videos we conduct experiments on subsets of Davis 2019 (Perazzi et al., 2016) and on FBMS (Ochs et al., 2014). For our experiments we process video annotations so that they are semantically consistent across videos in order to better measure our network's understanding of semantics. We use COCO Stuff (Lin et al., 2014) as a source of labeled images, which has many overlapping classes with Davis and FBMS.

Davis is a video object segmentation dataset with foreground object mask annotations for 60 training and 30 validation videos. We focus on 10 classes in common with COCO that appear in both training and validation splits. Having an object appear in the training set allows us to analyze how its object representation changes when using our method: any difference in performance results from changes in the object representations as opposed to the model's ability to generalize to unseen classes. A breakdown of the number of video frames in each of the shared classes in Davis can be found in Appendix Table 8.

FBMS contains 59 video sequences with 353 annotated training frames and 367 annotated test frames across 19 semantic classes. As an additional source of unlabeled video data that often features motion blur, low lighting, and low resolution, we use YouTube-BoundingBoxes (YTBB) (Real et al., 2017). It features bounding box annotations for 23 objects in ≈380,000 YouTube videos. To decode videos we use 30 fps and skip unavailable videos.

## 4.2 IMPLEMENTATION

To analyze the domain shift between images and videos, we train segmentation networks jointly on images and videos. We consider two types of segmentation backbones: CNNs and Transformers. For our CNN we use Deeplabv3 (Chen et al., 2017) with a ResNet-101 backbone initialized from COCO-pretrained weights. The domain discriminator consists of a 2D convolution, a fully connected layer, dropout, ReLU and the classification layer. For our 2D Transformer we use the Swin-T Transformer backbone with a UperNet decoder (Liu et al., 2021a) with patch size (4,4) and window size (7,7), initialized from ADE20k-pretrained weights. The discriminator is implemented as a sequence of alternating linear and leaky ReLU layers. When the discriminator is placed at the end of the encoder (global treatment) the tokens are concatenated and spatial dimensions flattened. At the patch embedding level (spatial treatment) each token's spatial dimensions are flattened and the token and batch dimensions are combined. Our spatiotemporal model is VideoSwin with a Swin-T backbone, patch size (1,4,4) and window size (4,7,7) and uses a PatchGAN discriminator (Isola et al., 2017) with four layers. VideoSwin is initialized with ADE20k weights replicated across the temporal dimension appropriately. The number of clusters in our adversarial loss is set to 11 and ablated in Table 6.

Table 1: Results on Davis from training a 2D CNN on Davis, COCO and YTBB. Adv indicates whether the model is trained with the adversarial loss. Labeled/Unlabeled columns indicate the label setting for the training set. mIOU is averaged over 10 classes plus background. The general adversarial loss boosts performance over the baseline trained on labeled images and no videos.

| Adv | Labeled | Unlabeled | mIOU | Δ |
|---|---|---|---|---|
| ✗ | COCO | – | 32.3 | – |
| ✓ | COCO | Davis | 36.9 | + 4.60 |
| ✓ | COCO | YTBB | 37.7 | + 5.40 |
| ✗ | Davis | – | 32.2 | – |
| ✗ | COCO+Davis | – | 46.2 | – |

## 5 RESULTS

### 5.1 2D CNNS ON DAVIS

We focus on the setting in which we have access to labeled images and unlabeled videos due to the challenging nature of collecting video annotations. To test the general adversarial loss' contribution to learning detailed semantic representations, we place a discriminator at the end of a 2D CNN encoder which acts on global spatial feature representations. We train on video frames from Davis, images from COCO and also experiment with training on random unlabeled YouTube videos from YTBB. Unlabeled frames are only used in the adversarial loss. Our baseline is the 2D CNN trained only on labeled images.

Table 1 shows our results on Davis measured using mean Jaccard index or Intersection over Union (IOU) over all semantic classes. We find that in our target setting with no access to labeled videos, the adversarial loss boosts the performance of the baseline CNN. The adversarial loss even boosts performance on Davis when using random unlabeled YouTube videos to train the discriminator. Our method outperforms the supervised setting in which the model is trained only on labeled videos. Training with the adversarial loss on images and unlabeled videos performs better than training on just labeled videos, highlighting the usefulness of our method in settings where labeled videos are difficult to collect and object diversity is limited.

To test whether our method can improve the performance of a VOS baseline model, we replace the CNN encoder of UNOVOST (Luiten et al., 2020) with our adversarially trained encoder and show this improves its performance on Davis in Table 7. Thus our method can boost performance of existing VOS models in the challenging unlabeled video setting.

### 5.2 TRANSFORMERS

In the following experiments we test whether the general and clustered adversarial losses boost the performance of a Transformer-based model trained only on labeled images. We use Swin as our 2D Transformer and VideoSwin with a spatiotemporal window size as our 3D Transformer.

**2D Transformer on Davis** We train Swin on videos from Davis and images from COCO and show results in Table 2. We find that applying the discriminator to spatial feature maps right after the patch embedding yields better performance than to the whole representation at the end of the encoder, showing that targeting spatial patterns is sufficient to reduce the domain gap. We test the performance of our clustered adversarial loss and show that clustering feature maps at the patch embedding level before applying the adversarial loss leads to a further boost in performance. Training with the adversarial loss improves performance in the unlabeled video setting as well as with access to labeled videos.

**VideoSwin on Davis** We explore how our method performs in a model that learns spatiotemporal features, namely, VideoSwin with image patch tokens and spatiotemporal window extending over the entire clip length. Table 3 shows the results of training this network using the general adversarial loss after the patch embedding. In the unlabeled video setting, the adversarial loss improves on the

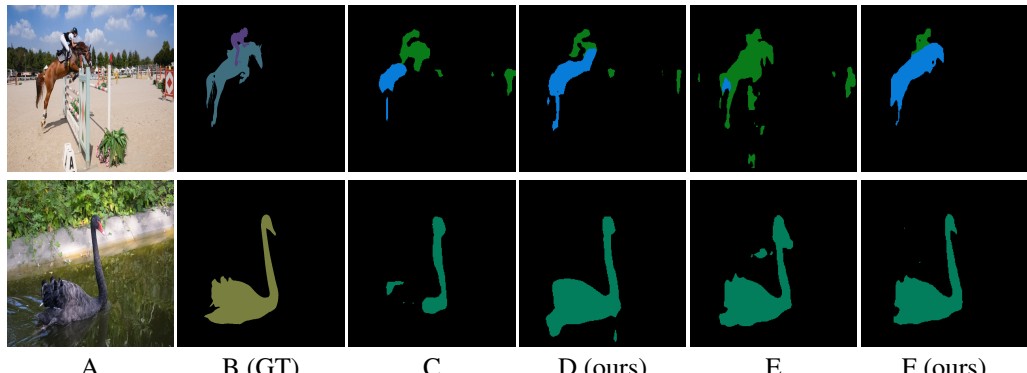

A        B (GT)        C        D (ours)        E        F (ours)

Figure 4: Qualitative results on Davis videos. *Column A:* Original frame. *Column B:* Ground truth. *Column C:* 2D CNN trained on COCO images. *Column D:* 2D CNN trained on labeled COCO and unlabeled Davis with general adversarial loss. *Column E:* Swin trained on COCO images. *Column F:* Swin trained with clustered adversarial loss on labeled COCO and unlabeled Davis frames. Swin trained with the clustered loss recovers fine-grained details and has a better separation of semantic classes than the model trained only on images.

Table 2: Results on Davis from training a 2D Transformer on Davis and COCO. Cluster indicates whether the clustered or general adversarial loss is used. The Spatial/Global column indicates the discriminator's location.

| Adv | Cluster | Spatial/Global | Labeled | Unlabeled | mIOU | Δ |
|---|---|---|---|---|---|---|
| ✗ | – | – | COCO | – | 29.98 | – |
| ✓ | ✗ | Spatial | COCO | Davis | 25.27 | - 4.71 |
| ✓ | ✗ | Spatial | COCO | Davis | 32.70 | + 2.72 |
| ✓ | ✓ | Spatial | COCO | Davis | 38.53 | + 8.55 |
| ✗ | – | – | Davis | – | 25.97 | – |
| ✗ | – | – | COCO + Davis | – | 29.01 | + 3.04 |
| ✓ | ✗ | Spatial | COCO + Davis | – | 28.15 | + 2.18 |
| ✓ | ✗ | Spatial | COCO + Davis | – | 32.55 | + 6.58 |
| ✓ | ✓ | Spatial | COCO + Davis | – | 38.27 | + 12.30 |

image-supervised model, showing that our method can be applied to spatiotemporal models that rely on image-pretrained weights.

**FBMS** We test whether our method achieves a boost in performance on a different video dataset FBMS and show results in Table 5. In the unlabeled video setting, the general adversarial loss improves over the model supervised by images. Because the annotations for FBMS do not include all instances of each class in every video, there is a discrepancy between the annotations for FBMS and COCO which we believe prevents the model from learning strong semantic class representations,

Table 3: Results on Davis of VideoSwin with the general adversarial loss at the patch embedding level.

| Adv | Labeled | Unlabeled | mIOU | Δ |
|---|---|---|---|---|
| ✗ | COCO | – | 19.35 | – |
| ✓ | COCO | Davis | 30.48 | + 11.13 |
| ✗ | Davis | – | 25.75 | – |
| ✗ | COCO+Davis | – | 32.46 | + 6.71 |
| ✓ | COCO+Davis | – | 33.65 | + 7.90 |

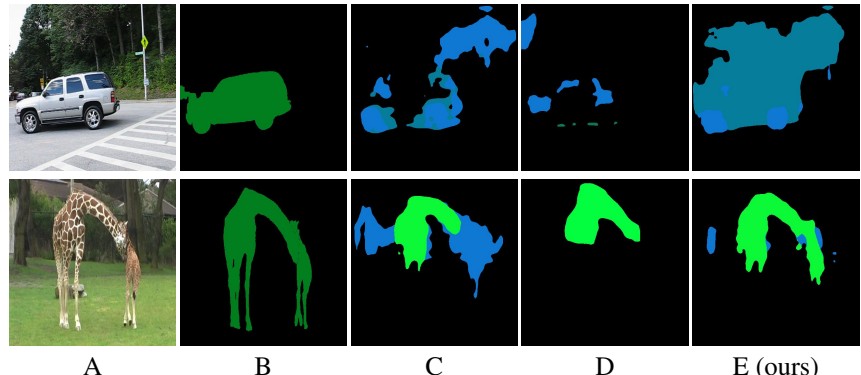

| A | B | C | D | E (ours) |

Figure 5: Qualitative results on FBMS videos. *Column A:* Original frame. *Column B:* Ground truth. *Column C:* Swin trained on FBMS video frames. *Column D* Swin trained on COCO images. *Column E:* Swin trained on labeled COCO and unlabeled FBMS data with the adversarial loss.

Table 4: Ablation study on whether an adversarial loss placed at the end of the encoder boosts improvement when training on labeled COCO images and unlabeled Pascal images. Adding unlabeled data from another image dataset does not significantly affect test IOU measured on COCO or Davis.

| Model | Adv | Labeled | Unlabeled | Test | mIOU |
|---|---|---|---|---|---|
| CNN | ✗ | COCO | – | Davis | 32.3 |
| CNN | ✓ | COCO | Pascal | Davis | 33.7 |
| CNN | ✗ | COCO | – | COCO | 44.5 |
| CNN | ✓ | COCO | Pascal | COCO | 45.0 |
| Transformer | ✗ | COCO | – | Davis | 10.35 |
| Transformer | ✓ | COCO | Pascal | Davis | 10.21 |
| Transformer | ✗ | COCO | – | COCO | 11.31 |
| Transformer | ✓ | COCO | Pascal | COCO | 10.10 |

leading the model to perform worse when trained with both labeled videos and images than training with our adversarial loss.

We show qualitative results of our method compared to the baselines on FBMS in Figure 5.2. When only training on video frames, the model is unable to fully represent the object and often under-segments it. When training on a large image dataset, the results are improved, and when training with the adversarial loss and unlabeled video frames, the model segments foreground objects better.

## 6 ABLATIONS

**Number of clusters** We conduct an ablation on the number of clusters $K$ used in our clustered adversarial loss, since increasing $K$ creates a trade-off with efficiency. In this ablation, we place the clustered adversarial loss after the patch embedding layer of a 2D Swin Transformer and train on labeled COCO images and unlabeled Davis frames. The results in Table 6 show that setting the number of clusters slightly above the minibatch size (11 clusters; minibatch size 8) performs best.

**Unlabeled Images** We conduct an ablation to test whether the performance boost with our adversarial loss is due to feature invariance in the image-video domains or invariance to dataset biases. To test this, we train a 2D model on labeled images from COCO and unlabeled images from the entire large-scale image segmentation dataset Pascal VOC (Everingham et al., 2010). The results in Table 4 show that using unlabeled images when training with the adversarial loss does not significantly improve results regardless of the backbone. Additionally the same observations hold when testing on COCO. We believe the adversarial loss' improvements are due to features becoming invariant to video artifacts, not dataset-specific biases.

Table 6: Ablation of number of clusters $K$ reported in mIOU on Davis.

| $K$ | 2 | 5 | 11 | 22 |
|---|---|---|---|---|
| mIOU | 45.57 | 47.27 | 47.97 | 47.25 |

Table 5: Results on FBMS of training a 2D Transformer with our general adversarial loss at the patch level.

| Adv | Labeled | Unlabeled | mIOU | $\Delta$ |
|---|---|---|---|---|
| ✗ | COCO | – | 33.98 | – |
| ✓ | COCO | FBMS | 34.47 | + 0.49 |

Table 7: Performance of UNOVOST (Luiten et al., 2020) on Davis with our adversarial training method.

| Adv | mIOU | $\Delta$ |
|---|---|---|
| ✗ | 26.49 | – |
| ✓ | 27.84 | +1.35 |

## 7 DISCUSSION

We hypothesize that due to different filter sensitivities, each image / video frame will have feature maps with many spatial structures, and that by including slightly more clusters than inputs, clusters include similar spatial patterns from multiple inputs. Because the clustered loss formulation does not explicitly rely on features specific to the image-video domain gap, it can be applied to different adaptation tasks. It could also be applied to prevent discriminative information loss during joint supervised training on multiple datasets. In this setting, when there is not significant overlap in semantic labels, missing information risks confusing the network and degrading performance.

In experiments with the adversarial loss within a 2D Transformer, we found placing it after the patch embedding layer increases performance over concatenating tokens at the end of the encoder and applying the loss over the concatenated feature. Feature maps at the beginning of the network retain fine-grained spatial details that likely contain more explicit domain information than features at the end of the encoder. This makes it easier for the discriminator to target video artifacts (such as motion blur at moving object boundaries). Placing the discriminator at the end of the encoder where features are abstract is also beneficial, so it may be useful to place the discriminator at intermediate blocks within the transformer, which we leave for future work.

Finally, we note that while spatiotemporal models have become critical for good performance in video classification, fully 3D end-to-end trainable networks have had limited success on video segmentation without relying on optical flow. We hypothesize that a lack of video annotations is a major contribution to this problem, and through our experiments using VideoSwin, we take a step towards developing a domain adaptation solution that relies only on image labels and abundant unlabeled videos.

## 8 CONCLUSION

In this paper we address a gap in the video representation learning literature where the focus has typically been on video classification rather than segmentation. We propose an unsupervised image-to-video domain adaptation method, in which a clustered adversarial loss allows a network to learn fine-grained semantic representations from labeled images and unlabeled videos. Similar to previous video models, we leverage image datasets to learn strong semantic features while using the adversarial loss to address the performance drop that ensues when applying these networks to videos. We conduct experiments for video segmentation to show that our method benefits learning detailed semantic representations for videos. We find that the general adversarial loss improves CNN and 2D/3D Transformer's performance and our clustered adversarial loss improves 2D Transformer performance.

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
