# A  APPENDIX

## A.1  DATASET DETAILS

We include a breakdown of the number of frames containing at least one pixel from a semantic class in the Davis dataset in Table 8. The final Davis training set consists of 3,401 frames and 1,558 frames for validation. When training with Davis the COCO training set has 75,377 frames and the validation set has 3,176 frames. When training with FBMS the COCO training set has 80,279 images of shared classes with FBMS.

## A.2  IMPLEMENTATION DETAILS

Our 2D CNNs are trained with batch size 8 and learning rate 0.001 and our 2D Transformer is trained with batch size 32 and learning rate 0.0032. Images and video frames are resized to 256×256 and normalized. VideoSwin is trained using batch size 16 and learning rate 0.0001. For VideoSwin we resize inputs to 512×512 and clip length 4 for videos, zero-padding single images to the same temporal resolution. We will publicly release our Pytorch code upon publication acceptance.

Loss coefficient $\mu$ is set to constant 1.0, which performed better than a ramp-up. The number of clusters is set to the number of classes in the video dataset, which is ablated in Table 6. On one GeForce GTX Titan X GPU, our 2D Transformer with the general adversarial loss takes 365 seconds to train an epoch on Davis with batch size 16 and 60.1 GFLOPS for inference. Under the same settings the clustered loss takes 1312 seconds and 60.0 GFLOPS.

| Class | Training Frames | Validation Frames |
|---|---|---|
| Backpack | 292 | 80 |
| Bicycle | 142 | 149 |
| Bird | 574 | 50 |
| Car | 394 | 217 |
| Cellphone | 182 | 94 |
| Dog | 277 | 241 |
| Horse | 60 | 50 |
| Motorcycle | 178 | 162 |
| Person | 4809 | 1621 |
| Surfboard | 345 | 50 |

Table 8: Number of frames in the training and validation splits for 10 Davis classes in common with COCO.