# OpenReview forum: "Unsupervised Image-to-Video Domain Adaptation for Fine-Grained Video Understanding"
_ICLR.cc/2024/Conference — ICLR 2024 Conference Withdrawn Submission_

### Official Review · Reviewer_stFh · 2023-10-30

**Soundness:** 2 fair
**Presentation:** 3 good
**Contribution:** 2 fair
**Rating:** 3
**Confidence:** 4

**Summary:**

This paper introduces a pre-training method for fine-grained video segmentation task. By leveraging the proposed clustered adversarial loss, it reduces the domain gap between image and video. The model is trained on labelled image data and unlabelled video data, and optimized by reducing the adversarial loss and pixel segmentation loss. The experimental results demonstrate the improvement from the proposed method over the baseline.

**Strengths:**

1. The paper is well written, which clearly demonstrates the motivation, methodology and experiments. The idea is easy to follow.
2. From the experimental results, the proposed method can improve the performance over the baseline method.

**Weaknesses:**

1. Motivation: The motivation of this paper is not well justified. The first sentence of the abstract and the first sentence of the introduction say video understanding model relies on image pre-trained model. However, there are many video self-supervised learning work that do not require image-based pre-training[1,2] . And there are also some works introducing the spatial features into the self-supervised learning[3]. More evidence will be useful to show why the proposed method is better than the video SSL works.


2. Methodology:
2.1 It is not novel to use adversarial loss to close the domain gap, which was explored in previous works[4,5].
2.2 Not all video frames include the artifacts mentioned in this paper, most of which are same with the regular image. In this case, how to make sure the proposed discriminator works as expected?
2.3 In section 3.2, pairs of image and video representation will be picked from one cluster. However, features will be put into the same cluster when they are close to each other. How to construct a balance number of feature pairs, given it's unsupervised?

3. Contribution: The experimental results demonstrate the gain over baseline result. However, the comparison with other state-of-the-art methods will be good to show the contribution of this work.


1.A Large-Scale Study on Unsupervised Spatiotemporal Representation Learning, CVPR21

2.Long-Short Temporal Contrastive Learning of Video Transformers, CVPR22

3.Contextualized Spatio-Temporal Contrastive Learning With Self-Supervision, CVPR22

4.Adversarial-Learned Loss for Domain Adaptation, AAAi20

5.Adversarial Discriminative Domain Adaptation, CVPR17

**Questions:**

1. This work leverage the labelled image data and unlabelled video data for the training. In practical, unlabelled video data is from the video object detection/segmentation datasets, but provide no annotation during the pre-training. Is this a hard requirement for this work? Can general video work?

2. The proposed idea should also work in other domain adaptation work, such as first personal view video vs. third personal view video. Is this work tried in other applications? which will be helpful to provide more insights and make this paper stronger.

3. In Table 2, the 2nd and 3rd in the upper half and the 3rd and 4th in the lower half are same. There should be global setting?

---

### Official Review · Reviewer_jnCk · 2023-10-31

**Soundness:** 2 fair
**Presentation:** 2 fair
**Contribution:** 2 fair
**Rating:** 3
**Confidence:** 4

**Summary:**

This paper aims to address the task of image-to-video semantic segmentation in an unsupervised setting. It proposes the use of DOMAIN ADVERSARIAL LOSS and CLUSTERED ADVERSARIAL LOSS, placed respectively in the deep and shallow layers of the network, to assist the network in learning video semantic segmentation with only image annotations.

**Strengths:**

The experiments were conducted on subsets of two video datasets, showing improvements compared to a self-constructed baseline.

**Weaknesses:**

The quality of this article is unsatisfactory, including the following issues and more:

1. Poor writing quality and logical coherence. This is evident in various aspects, such as the mismatch between the title's "FINE-GRAINED VIDEO UNDERSTANDING" and the actual focus on video semantic segmentation. Additionally, the abrupt logic in the first two sentences of the introduction lacks thorough analysis and proper referencing. Furthermore, the paper emphasizes video artifacts, such as motion blur, low lighting, and low resolution, as reasons for the difficulty in efficiently utilizing image pretraining for video segmentation. However, this claim lacks supporting citations and in-depth theoretical or experimental analysis, while relying heavily on unsubstantiated hypotheses. The meaning of the right half of Figure 2 is unclear, and the organization of content in Chapter 5 is illogical.

2. This article only uses a subset of the complete dataset and lacks performance comparisons with state-of-the-art models, making it difficult to demonstrate the effectiveness of the proposed methods. Furthermore, there are some peculiar experimental results, such as the unsupervised result outperforming the supervised result in the 5th row of Table 2.

3. The references in this article do not conform to the standard format, with some formally published works being cited as arXiv versions.

4. Typos: The ambiguous symbols: θ and La in Formula (5)

**Questions:**

Please see my comments in Weaknesses.

---

### Official Review · Reviewer_g8Ho · 2023-10-31

**Soundness:** 2 fair
**Presentation:** 1 poor
**Contribution:** 1 poor
**Rating:** 1
**Confidence:** 4

**Summary:**

Proposes an approach for domain adaptive semantic segmentation that aligns representations for labeled images and unlabeled videos using a clustered adversarial loss. Presents results and demonstrates performance gains on several benchmarks over baselines.

**Strengths:**

– The paper studies an interesting and understudied unsupervised domain adaptation (UDA) subproblem – that of leveraging unlabeled videos and labeled images. As the authors point out, this subproblem is particularly relevant for dense prediction tasks.

– The problem is well-motivated

– The experimental section is fairly comprehensive, with results on several architectures and datasets

**Weaknesses:**

– The paper does not compare to the wide array of baselines from the UDA literature, which I imagine would be straightforward to benchmark for this setting. While it does include some reasonable ablations, that is insufficient to conclude that the proposed method is sufficiently performant.

– The experimental procedure is inconsistent, for eg. using the cluster adversarial loss in Table 2 whereas the general adversarial loss in Tables 3, 5, bringing the efficacy of the paper’s main technical contribution (clustered adversarial loss) into question. The paper does not provide a reason for this inconsistency.

– Sec 3.2 motivates the clustered adversarial loss by saying that “image and video datasets may be labeled with different semantic classes and we seek to take advantage of all labeled data regardless of annotation consistency”. However, in a setting with such an inconsistent annotation protocol (FBMS), it does not use clustering altogether, which is somewhat contradictory.

– The experimental section is hard to follow and altogether confusingly structured – rows are grayed out without explaining what that indicates, and improvements from qualitative results are hard to interpret. While several experiments are conducted, both their motivation and takeaways are frequently lacking.

– The proposed method shares considerable similarities with prior work in UDA for semantic segmentation. For example Tsai et al., [A], CVPR 2018 also propose a multi-level domain-adversarial approach for CNNs. While the proposed method considers a different setting (image-to-video adaptation) and additionally performs clustering before adversarial alignment, the technical contribution is therefore rather limited.

– Typos: In Table 2, rows 2-3 and 7-8 are identical. Should it say “global” in Rows 2 and 7 instead?

[A] Tsai et al., Learning to Adapt Structured Output Space for Semantic Segmentation, CVPR 2018

**Questions:**

Please address the concerns raised in the weaknesses section above.

---

### Official Review · Reviewer_RLFw · 2023-11-01

**Soundness:** 2 fair
**Presentation:** 1 poor
**Contribution:** 2 fair
**Rating:** 3
**Confidence:** 4

**Summary:**

The paper addresses domain gap between images and videos, especially for the task of video segmentation. It modifies the widely used adversarial loss for domain adaptation with clustering so as to retain overall spatial structures across images and videos while discarding image or video domain-specific features. The proposed method itself seems to make sense, but the writing/presentation should be  improved and additional experiments are needed for showing the validity of the proposed method.

**Strengths:**

+ I acknowledge the need for addressing the domain gap between images and videos, as shown in Figure 1, as well as more fine-grained image-to-video adaptation approaches for video segmentation.
+ The proposed method, using an adversarial loss with clustering, makes sense, and is simple yet quite effective, which is proven through experiments.
+ The ablation study to test whether the performance boost is due to learning invariance to image-video domains or dataset biases clearly shows the adversarial loss works as it is expected.

**Weaknesses:**

- It seems that the main contribution of this paper is the modified adversarial objective using clustering. However, the clustered adversarial loss is validated only in a single experimental setting: training a 2D Transformer on Davis and COCO. Unless there are any specific reasons the authors show the results of the clustered adversarial loss only in this setting, they should provide results in more diverse settings, such as training a 2D Transformer on FBMS and COCO or training with a 3D Transformer/2D CNN.
- The writing and presentation should be improved. First, the authors should elaborate more on the underlying intuition or motivation behind the clustered adversarial loss. Section 3.2 just describes how it works, but does not explain why it can remove domain-specific features while retaining overall spatial semantic information. More importantly, Table 2, which contains the most important experimental results, has multiple identical rows (Row 2-3 in the first half and Row 3-4 in the second half). This makes it very hard to understand the results.

**Questions:**

Please respond to the weaknesses I mentioned above.

Minor comments: there are many typos in equations, e.g., equation (2) and (3) (signs), equation (4) and (5) (subscripts). Please fix these in the final draft.

---

### Author Response · Authors · 2023-11-22
**Clarification on Motivation and Novelty**

We thank the reviewers for their suggestions.

We first clarify the motivation of our work: that image-to-video domain adaptation is an important understudied problem for pixel-wise classification in videos because: 1. collecting pixel-wise video labels is difficult 2. existing UDA methods and video SSL methods learn representations that are too coarse for pixel-wise classification in videos.

Our main claim is that using an adversarial loss to address image-to-video domain adaptation for pixel-wise video classification has not been explored in the literature. In particular, our clustered adversarial loss is novel. To address any confusion around the clustered adversarial loss, we elaborate on its underlying intuition and motivation: We want to devise a domain adaptation method for the image-to-video domain gap by targeting semantic representations within a pixel-wise classification network. However, certain classes are well represented within the dataset, so applying a typical adversarial loss to the features within the network risks losing discriminative semantic information for rare classes. Instead, we apply a domain adversarial loss to features that predominantly contain information for a single class. We implement this by clustering features in an unsupervised way, then applying an adversarial loss within each cluster. Table 2 shows the efficacy of this method.

We will revise the paper to 1. Choose a well-studied baseline and 2. Show the efficacy of our clustered loss in further settings.

We withdraw the paper to give it an opportunity to be reviewed by another conference.